# Tumor Necrosis Factor Receptor 2 (TNFR2): An Emerging Target in Cancer Therapy

**DOI:** 10.3390/cancers14112603

**Published:** 2022-05-25

**Authors:** Juliane Medler, Kirstin Kucka, Harald Wajant

**Affiliations:** Division of Molecular Internal Medicine, Department of Internal Medicine II, University Hospital Würzburg, Auvera Haus, Grombühlstrasse 12, 97080 Würzburg, Germany; kums_j@ukw.de (J.M.); kucka_k@ukw.de (K.K.)

**Keywords:** NFkappaB, regulatory T-cell (Treg), tumor necrosis factor (TNF), TNF receptor 2 (TNFR2), TNF receptor associated factor 1 and 2 (TRAF1, TRAF2)

## Abstract

**Simple Summary:**

Tumor necrosis factor (TNF) receptor-2 (TNFR2) affects tumor development and metastasis in several ways. TNFR2 promotes tumor immune escape by virtue of its ability to stimulate various immune suppressive cell types, e.g., regulatory T-cells (Tregs) and myeloid-derived suppressor cells (MDSCs) and can act as an oncogene. However, TNFR2 also elicits antitumoral activities by costimulation of cytotoxic T-cells. Accordingly, antagonists and agonists targeting TNFR2 have been preclinically evaluated for tumor therapy and have demonstrated anti-tumor activity in preclinical studies. In this review, we summarize the most important TNFR2-related findings regarding tumor biology and cancer therapy and especially discuss the mode of action of currently used agonists and antagonists of TNFR2.

**Abstract:**

Despite the great success of TNF blockers in the treatment of autoimmune diseases and the identification of TNF as a factor that influences the development of tumors in many ways, the role of TNFR2 in tumor biology and its potential suitability as a therapeutic target in cancer therapy have long been underestimated. This has been fundamentally changed with the identification of TNFR2 as a regulatory T-cell (Treg)-stimulating factor and the general clinical breakthrough of immunotherapeutic approaches. However, considering TNFR2 as a sole immunosuppressive factor in the tumor microenvironment does not go far enough. TNFR2 can also co-stimulate CD8^+^ T-cells, sensitize some immune and tumor cells to the cytotoxic effects of TNFR1 and/or acts as an oncogene. In view of the wide range of cancer-associated TNFR2 activities, it is not surprising that both antagonists and agonists of TNFR2 are considered for tumor therapy and have indeed shown overwhelming anti-tumor activity in preclinical studies. Based on a brief summary of TNFR2 signaling and the immunoregulatory functions of TNFR2, we discuss here the main preclinical findings and insights gained with TNFR2 agonists and antagonists. In particular, we address the question of which TNFR2-associated molecular and cellular mechanisms underlie the observed anti-tumoral activities of TNFR2 agonists and antagonists.

## 1. Introduction

Tumor necrosis factor (TNF) receptor 2 (TNFR2) is a type I transmembrane protein and a prototypic member of the TNF receptor superfamily (TNFRSF) [1,2]. TNFR2 belongs to the TRAF (TNF-receptor-associated factor) -interacting subgroup of the TNFRSF and mediates pro-inflammatory effects, but can also stimulate strong anti-inflammatory activities [3]. TNFR2 expression is typically high in myeloid cells but is also found in certain T- and B-cell subsets and a few non-immune cells such as endothelial cells, glial cells and cardiomyocytes. For example, macrophages and regulatory T-cells (Tregs) constitutively express TNFR2 [3]. Expression of TNFR2 is furthermore upregulated after TCR (T-cell receptor) stimulation in effector T-cells (Teff).

Members of the TNFRSF are characterized by having one to six cysteine rich domains (CRDs) in their extracellular part, TNFR2 contains four of these CRDs [2] (Figure 1). The first N-terminally located CRD of TNFR2 (CRD1) acts as a PLAD (pre-ligand binding assembly domain) mediating low-affinity (compared to ligand affinity) autoassociation of receptor molecules. This suggests that TNFR2 is present in the plasma membrane in a dynamic equilibrium of inactive mono- and di-/trimeric receptor species. CRD2 and CRD3 are involved in ligand binding [4] (Figure 1). A specific function of CRD4 has not been identified yet. CRD4 of TNFR2 is separated from the transmembrane domain by a stalk region which antagonizes PLAD-mediated receptor clustering [5] (Figure 1).

The naturally occurring ligands of TNFR2 are membrane-bound TNF (memTNF), soluble TNF (sTNF), homotrimeric lymphotoxin α (LTα) and soluble and membrane-bound LTα_2_β heterotrimers [1,6]. TNF is a type II transmembrane protein of the TNF superfamily (TNFSF) and forms stable homotrimers via the highly conserved extracellular TNF homology domain (THD) [1,2,7]. TNF is mainly produced by immune cells, especially by activated T-cells and macrophages after contact with pathogen-associated molecular patterns (PAMPs) or danger-associated molecular patterns (DAMPs) [8]. In view of its outstanding role in autoimmune diseases, sepsis and fibrosis, it is not surprising that TNF is the target of several approved and widely clinically used biologicals [9]. The memTNF variant can be converted to sTNF by the matrix metalloprotease TNF-alpha converting enzyme (TACE) [10,11]. Soluble TNF and homotrimeric LTα, which only occurs as a soluble molecule, do not activate robust TNFR2 signaling despite high-affinity binding [12]. TNF and LTα_2_ß, in their membrane-bound form, however, trigger TNFR2 signaling [6,12]. LTα binds additionally herpes virus entry mediator (HVEM), another TRAF-interacting receptor of the TNFRSF [13]. Like both TNF variants, LTα is also a ligand for TNFR1, a receptor of the death receptor subgroup of the TNFRSF [1,2]. TNFR1 is ubiquitously expressed and is activated by soluble and membrane-bound ligand molecules. TNFR1 stimulates strong pro-inflammatory signaling, particularly via the classical NFκB pathway, but can also trigger apoptosis and necroptosis [14].

The differential ability of sTNF and memTNF to trigger TNFR2 signaling seems to reflect the differential clustering capacity of sTNF- and memTNF-liganded TNFR2 complexes [15]. The PLAD of TNFR2 not only enables the formation of receptor complexes which display higher ligand affinity, but may also efficiently promote the clustering of memTNF-liganded TNFR2 complexes in the case of the extremely high local concentration of these complexes in the cell-to-cell contact zone formed between memTNF and TNFR2-expressing cells. The low affinity of the PLAD of TNFR2, however, seems not to be sufficient to promote the clustering of sTNF-liganded TNFR2 complexes against the repulsive forces originating from the stalk region [5]. In contrast, TNFR1 lacks this repulsive stalk and already clusters after binding of sTNF has recently been demonstrated by photoactivated localization microscopy (PALM) and modeling approaches [16,17,18].

## 2. TNFR2 Signaling

The main signaling pathways activated by TNFR2 stimulation are the classical (or canonical) and the alternative (or non-canonical) NFκB (nuclear factor ‘kappa-light-chain-enhancer’ of activated B-cells) pathway. TNFR2 is also linked by poorly understood signaling cascades in a cell-type-specific manner to the endothelial/epithelial protein tyrosine kinase (Etk), phosphatidylinositol 3-kinase (PI3K) and its effector protein kinase B (Akt) [19,20,21]. The NFκB transcription factor family comprises mammals’ five structurally related proteins (RelA/p65, RelB, cRel, p105/p50 and p100/52) which form various homo- or heterodimers with different transcriptional properties [22]. The p50-p65 dimer is the main target of the classical NFκB pathway. In unstimulated cells, this NFκB dimer is kept in the cytoplasm in an inactive complex by interaction with inhibitory IκB (inhibitor of the nuclear factor kappa B) proteins which mask the nuclear localization sequence (NLS) of the p50/p65 dimer [22].

The key activating event in the classical NFκB pathway is the stimuli-induced proteasomal degradation of the aforementioned inhibitory IκB proteins. IκB proteins are marked for degradation in a stimuli-dependent fashion by phosphorylation by the IKK (IκB kinase) complex, which in turn is itself activated by phosphorylation. In the case of TNFR2, the stimulatory phosphorylation of the IKK complex is linked to activated receptor clusters by the recruitment of TRAF2 and the TRAF2-interacting E3 ligases cIAP1 (cellular inhibitor of apoptosis protein-1) and cIAP2 (Figure 2) [23]. The cIAPs K63 ubiquitinate components of the TNFR2 signaling complex, presumably mainly TRAF2, to enable the recruitment of the so called LUBAC (linear ubiquitin chain assembly complex), which modifies components of the TNFR2 signaling complex with M1-polyubiquitin chains [24,25].

The mechanisms of how M1- and K63-ubiquitination of TNFR2 signaling complex components trigger IKK complex phosphorylation/activation have not been investigated yet. However, it is tempting to speculate that this takes place via similar mechanisms as in the case of TNFR1, where K63- and M1-polyubiquitin chains act as docking sites for the IKK complex and K63-polyubiquitin chains also recruit the IKK-phosphorylating TAK1-binding-protein-2 (TAB2)/TAB3-transforming growth factor β (TGFβ)-activated kinase 1 (TAK1) complex [22,26,27]. After the degradation of the IκB proteins, the NFκB dimers translocate into the nucleus and induce transcription of a variety of genes controlling inflammation, immunity, cell proliferation, differentiation and survival [22,28,29].

Recruitment of TRAF2 and the cIAPs to TNFR2 is not only necessary to stimulate the classical NFκB pathway, but can also result in the inhibition of events requiring the cytoplasmic pool of TRAF2-cIAP1/2 complexes. For example, in unstimulated cells, the NFκB-inducing kinase (NIK) is constitutively K48-ubiquitinated by cIAP1 and cIAP2 in a TRAF2- and TRAF3-dependent manner and is subsequently degraded by the proteasome. Interference with this constitutively active inhibitory mechanism results in NIK accumulation and NIK-mediated phosphorylation of IKK1. The latter then phosphorylates p100-RelB, which results in the limited processing of p100 to p52 and the nuclear translocation of p52-RelB NFκB dimers [22,30,31,32]. The recruitment of TRAF2-cIAP1 and TRAF-cIAP2 complexes by TNFR2 reduces the availability of these complexes in the cytoplasm and accordingly inhibits NIK degradation and stimulates the alternative sequence of NFκB activation [33,34].

The opposing roles of TRAF2-cIAP1/2 complexes in classical and alternative NFκB signaling are worth highlighting. For the activation of the non-canonical NFκB pathway, the sole recruitment of the TRAF2-cIAP1/2 complexes to TNFR2 is sufficient because this limits the cytosolic availability of these complexes. Activation of classical NFκB signaling also needs the interaction of two TRAF2-cIAP1/2 complexes because a homotrimeric TRAF2 (or heterotrimeric TRAF1-TRAF2) molecule binds only one cIAP1 or cIAP2 molecule, and transactivation of two cIAP molecules is necessary to unleash their E3 ligase activity, which has a crucial role in classical NFκB activation [35,36,37].

The availability of cytoplasmic TRAF2-cIAP1/2 complexes is also a crucial factor that determines the quality of TNFR1 signaling. If there is reduced availability of cytoplasmic TRAF2-cIAP1/2 complexes, e.g., due to stimulation of TNFR2 or other TRAF2-interacting TNFRSF receptors, such as Fn14, CD30 or CD40, the ability of TNFR1 to induce cell death is strongly enhanced [33,38,39,40,41,42]. The reasons for this sensitizing effect are complex and include reduced TNFR1-induced, TRAF2-cIAP1/2-mediated production of survival factors controlled by the classical NFκB pathway, such as FLIP and cIAP2, but also diminished inhibition of caspase-8 maturation and necroptotic receptor-interacting serine/threonine-protein kinase-1 (RIPK1)-RIPK3 interaction by cIAPs. It is worth mentioning that the sensitizing effect on TNFR1-induced cell death can cooperate with the induction of TNF via the alternative NFκB pathway by TNFR2 or TNFR2-related TNFRs [40,43]. This explains, why TNFR2 stimulation can lead to cell-death induction in some cell types, e.g., macrophages, although TNFR2 itself is not a death receptor [8,44,45].

## 3. TNFR2 in Immune Regulation

Several types of immune cells express TNFR2, including regulatory T-cells (Tregs), regulatory B-cells (Bregs), myeloid-derived suppressor cells (MDSCs) and cytotoxic CD8^+^ T effector cells (Teffs) [3]. Human and mouse regulatory T-cells are characterized by the expression of CD4, the IL2 receptor subunit CD25, and Foxp3, which is the major transcription factor responsible for Treg identity [46,47]. Tregs make up around 10% of the peripheral CD4^+^ T-cells, and 30–40% of peripheral-activated Tregs express TNFR2 [48,49]. The main function of Tregs is to tune down the activity of activated T-cells to prevent autoimmune diseases and excessive inflammation. Tregs, however, can also be exploited by tumor cells for immune escape [50]. Tregs expressing high levels of TNFR2 show maximal suppressive activity [48]. Contrary, TNFR2 low/negative Tregs, which make up 60–70% of classical Tregs, display less suppressive activity [47,48,51]. TNFR2 induces activation of the classical and the alternative NFκB signaling pathway. The same signaling pathways are activated by 4-1BB, CD27, OX40 and GITR, which are also TRAF2-interacting receptors of the TNFRSF and which have also been involved in the development and survival of Tregs [52] (Figure 3).

As early as 2009, it was detected that enhanced NFκB activity increases the number of Foxp3^+^ cells in the thymus by promoting T-cell receptor-induced cRel-mediated transcription of Foxp3 [53]. Furthermore, Grinberg-Bleyer et al. uncovered a pivotal role of the canonical NFκB subunit cRel for the generation and maintenance of activated Tregs [54]. Accordingly, deficiency of cRel or p65 in Tregs has been associated with impaired protection against T-cell transfer-induced colitis [55]. These results are supported by the finding of Lubrano di Ricco et al. that anti-CD3 and anti-CD28 treated murine Tregs co-stimulated with a TNFR2 agonist display increased RelA-dependent proliferation and survival [56]. Furthermore, mice with Tregs that lack p100 expression show impaired suppressive Treg activity and massive inflammation [57]. TNFR2-mediated expansion and maintenance of Tregs come along with the maintenance of CpG demethylation of the Foxp3 promoter [58]. Another study found furthermore that an agonistic TNFR2-specific antibody induces the expression of the histone methyltransferase *EZH2* [59], a target of NFκB transcriptions factors, which forms a complex with Foxp3 in activated Tregs that is important for the suppressive activity of this cell type [59,60,61]. In sum, these results suggest that TNFR2 and related TNFRs act in Tregs, as known for other cell types, via the classical and the alternative NFκB pathway. It is worth mentioning that there is evidence from tumor-associated macrophages that TRAF2 also promotes proteasomal degradation of cRel [62]. Thus, by virtue of its ability to deplete cytosolic TRAF2 pools, as documented in macrophages, TNFR2 may also increase cRel expression at the posttranscriptional level in Tregs. Furthermore, TNFR2 supports Treg proliferation and stability of Treg identity by inducing glycolysis via the PI3-kinase (PI3K)-Akt-mammalian target of rapamycin (mTOR) signaling pathway [63]. However, how the PI3K-Akt-mTOR axis is linked in detail to TNFR2 in Tregs, but also in general, remains to be clarified.

Not only the prototypic CD25^+^-Foxp3^+^-CD4^+^ Tregs express TNFR2, but also CD8^+^Foxp3^+^ Tregs can be TNFR2 positive. The main fraction of CD8^+^Foxp3^+^ Tregs generated from peripheral blood mononuclear cells (PBMCs) with anti-CD3 monoclonal antibodies are CD25 and TNFR2 positive, but independent of CD25 expression, TNFR2 expression marks the most suppressing subset of this Treg type [64]. An even stronger inhibition of effector-cell proliferation can be detected by TNFR2 and programmed cell death ligand 1 (PDL1) positive CD8^+^FoxP3^+^ Tregs [64,65]. Notably, it has been reported that TNF inhibition by neutralizing monoclonal antibodies such as Infliximab and Adalimumab leads to increased blood Tregs in patients with rheumatoid arthritis (RA) [66], and another study showed that exogenous sTNF reduces the inhibitory effect of Tregs from HBV (hepatitis B virus) patients [67]. The observations made in these reports might reflect a Treg-inhibitory activity of the sTNF-TNFR1 axis, e.g., TNFR1-induced cell death.

Besides the Tregs, the majority of CD4^+^ T-cells are TNFR2 negative. Only a small fraction of mouse CD4^+^ T effector cells (Teffs) upregulate TNFR2 expression in response to TCR stimulation. This results in higher proliferation, increasing production of effector cytokines such as IFNγ and stronger resistance to Treg-mediated inhibition [47,51,68].

TNFR2 is also highly expressed on naïve primary CD8^+^ T lymphocytes and initially promotes their survival, differentiation and proliferation after TCR activation [68,69,70] (Figure 3). Furthermore, 4-1BBL^+^ B-cells, which are enriched in elderly humans and mice, were found to upregulate memTNF and to stimulate CD8^+^ T-cells via TNFR2 [71,72]. Interestingly, the latter come along with enhanced anti-tumor activity [72,73]. However, TNFR2 can also act as a negative regulator of activated CD8^+^ T-cells by contributing to activation-induced cell death (AICD). TNFR2-deficient CD8^+^ Teffs are highly resistant to AICD displaying enhanced expression of pro-survival signaling molecules, such as as Bcl-2, survivin and CD127, and reduced sensitivity for TNFR1-induced cell death [74,75]. The latter could reflect an increase in freely available cytoplasmic TRAF2-cIAP1/2 complexes for TNFR1 in TNF-stimulated TNFR2-knockout cells due to the lack of TNFR2-dependent sequestration of TRAF2-cIAP1/2 complexes, as discussed above. However, there is also evidence from human T-cells that TNFR2 mediates AICD by triggering production of reactive oxygen species (ROS) [76]. TNFR2-activated CD8^+^ Teffs increase memTNF expression which in turn, however, can promote Treg proliferation via TNFR2 [77]. Thus, memTNF-TNFR2 axis seems to prevent excessive immune reactions by supporting AICD activity in CD8^+^ T-cells and stimulation of CD4^+^ and CD8^+^ Tregs.

MDSCs are a rather heterogeneous population of myeloid cells characterized by the expression of CD11b and Gr1. They are negative regulators of the immune responses and can be exploited by tumors to escape from anti-tumor immunity [78]. MDSCs of TNF- and TNFR2-deficient mice have lower immune-suppressive capacities compared to MDSCs derived from wildtype mice [79,80]. MemTNF/TNFR2-activated MDSCs accordingly display enhanced expression of arginase-1, TGFβ and interleukin-10 (IL10) and increase NO production [80,81]. Furthermore, TNFR2 signaling promotes MDSC survival through upregulation of the cellular FLICE inhibitory protein (cFLIP), a powerful inhibitor of death-receptor-induced apoptosis [79]. Notably, memTNF seems also to act as an effector molecule of MDSCs through stimulation of TNFR2^+^CD4^+^ T-cells [82].

Three other cell populations with immune-regulatory activity involving the TNF-TNFR2 axis are endothelial progenitor cells (EPCs), group 2 innate lymphoid cells (ILC2s) and mesenchymal stem cells (MSCs). Naserian et al. recently described the ability of EPCs to suppress T-cell proliferation under crucial involvement of TNF and TNFR2 [83], and Hurrell et al. identified the TNF-TNFR2 axis as a mediator of pulmonary ILC2 survival and function, and also as a crucial factor for ILC2-mediated airway hyperreactivity [84]. The latter highlights TNFR2 as a potential therapeutic target in ILC2-dependent asthma. TNFR2 has been crucially implicated in MSCs in the ability of this cell type to inhibit proliferation and activation of different populations of T-cells [85,86], not least by tipping the balance from the production of proinflammatory cytokines to anti-inflammatory cytokines, such as IL-10 and TGFβ, which promote conversion of CD3^+^CD25^-^ conventional T-cells to CD4^+^Foxp3^+^ Tregs [85,87].

Given the various important functions of TNFR2 in a variety of immune cells, but also in certain non-immune cells, it is not surprising that TNFR2 also plays an important role in many types of cancer.

## 4. TNFR2 in Cancer

Depending on the cell type on which TNFR2 is expressed (Figure 3), it drives pro- or anti-inflammatory activities, as described in the previous section. This is also evident in tumor biology. The aberrant expression of TNFR2 or enrichment of TNFR2-positive immune-suppressive cell types, such as Tregs and MDSCs, in the tumor microenvironment (TME) may contribute to the evasion of tumor cells from the immune system. In contrast, TNFR2 provides also anti-tumoral activities when it, for example, co-stimulates tumor-specific CTLs.

### 4.1. TNFR2 in Tumor Immune Escape

Tumor escape from the immune system is a central step in tumor development. Loss of antigenicity, loss of immunogenicity and an immunosuppressive microenvironment are the three main mechanisms of tumor immune escape [88]. Thus far, the relevance of TNFR2 for immune escape has mainly been attributed to the beneficial effects of TNFR2 on immunosuppressive tumor-infiltrating Tregs and MDSCs [89]. Furthermore, earlier studies demonstrating a tumor-promoting function of TNF and IL10-producing regulatory B-cells in DMBA/TPA-induced skin carcinogenesis and the recent finding that TNFR2 stimulates IL10 production of regulatory B-cells open the possibility that TNFR2 engaged Bregs also contributes to the anti-inflammatory pro-tumoral effects of TNFR2 [90,91].

Tumor-infiltrating regulatory T-cells are, with high frequency, strongly positive for TNFR2 [92]. For example, in breast cancer, acute myeloid leukemia (AML) and lung cancer, the highest TNFR2 expression levels were found on Foxp3^+^CD25^+^CD4^+^ Tregs [93,94,95,96]. Interestingly, there is evidence that chemotherapy affects the TNFR2^+^ Treg pool in triple-negative breast tumors more severely than the infiltrating CD8^+^ T-cells, so in this special treatment situation, the anti-tumoral TNFR2 activities prevail [97].

As discussed before, TNFR2^+^ Tregs represent the most suppressive fraction of Foxp3^+^ cells and activation of TNFR2 on Tregs by memTNF comes along with increased proliferation and phenotypic stability [59,98]. Soluble TNFR2 is significantly higher in samples of malignant epithelial ovarian cancer (EOC) than in corresponding benign neoplasias and is associated with tumor differentiation [99]. Moreover, TNFR2^+^ Tregs are abundant in the ascites of ovarian tumor patients and have higher suppressive activity than peripheral blood TNFR2^+^ Tregs [100]. Notably, antagonistic TNFR2-specific antibodies trigger cell death more potently in Tregs isolated from the ascites of ovarian cancer patients than in Tregs of healthy donors [101]. Furthermore, a study of tumor ascites from patients with EOC revealed that high levels of the pro-inflammatory cytokine IL6 were present in the analyzed ascites. Culturing T-cells with EOC ascites leads to an increased TNFR2-expression on all analyzed T-cell subsets and an increased ratio of Tregs/Teffs. In addition, those cultured Tregs express higher levels of immunosuppressive molecules such as programmed cell death ligand-1 (PDL1) and cytotoxic T-lymphocyte-associated protein 4 (CTLA4) [102], both molecules with an overwhelming importance for tumors to evade the immune system. Increased expression of CTLA4 and PDL1 was also reported for TNFR2^+^ Tregs derived of lung cancer patients [95,96]. Like IL6, TNF can also upregulate PDL1 expression. Lim and colleagues found out that TNF stabilizes the expression of PDL1 on cancer cells [103]. Mechanistically, TNF activates NFκB signaling, which induces COP9 signalosome 5 (CSN5) expression. CSN5 itself inhibits ubiquitination and subsequent degradation of PDL1. However, whether this mechanism is also of relevance in Tregs is unclear.

The concentrations of soluble TNFR1 and TNFR2 and TNF are significantly increased in sera of AML patients [104,105]. Furthermore, various mouse models of AML with distinct genetic abnormalities showed that autocrine TNF production and TNFR activation by AML cells, especially leukemia-initiating cells, make a crucial contribution to tumor progression through NFκB and JNK-mediated survival signaling [105,106,107], which might antagonize tonic TNF-driven necroptotic signaling [108]. In a FLT3-ITD driven model of AML, however, TNFR or RIPK3 knockout also resulted in enhanced leukemogenesis [109]. These seemingly contrasting results possibly reflect differences in the way the balances between cytotoxic and survival pathways in general has been adjusted in the various models due to genetic factors or differences in the micromilieu. It is, however, also possible that subtle, model-specific differences in the activity of distinct TNF-TNFR1-TNFR2 signaling network axes (sTNF versus memTNF, thus TNFR1 versus TNFR2 activity, induction of endogenous TNF via TNFR1 and/or TNFR2, TNFR2-mediated TRAF2 depletion, etc.) create these strikingly opposing net effects. In this respect, it is worth mentioning that the RNA N6-methyladenosine reader enzyme YTH N6-methyladenosine RNA binding protein 2 (YTHDF2) promotes AML and reduces cytotoxic TNF sensitivity of preleukemic cells by suppressing TNFR2 expression [110,111]. Indeed, in accordance with the idea that this mirrors the ability of TNFR2 to sensitize for TNFR1-induced cell death signaling by restricting the availability of TRAF2-cIAP/1/2 complexes (see introduction), SMAC mimetics which deplete cIAPs by triggering their proteasomal autodegradation also sensitizes AML cells for autocrine TNF-mediated necroptosis [112].

Tregs have been shown to possess additional mechanisms for inhibiting the induction of inflammatory pathways beyond the secretion/expression of anti-inflammatory cytokines, e.g., the release of soluble TNFR2 (sTNFR2). Shedding of the TNFR2 ectodomain, similarly to membrane TNF processing, is mediated by TACE/ADAM17 [113] and is often enhanced under inflammatory circumstances. In particular, sTNFR2 is released from CD4^+^ T-cells after stimulation with TNF or agonistic TNFR2-specific antibodies [114] while stimulation of TNFR1, but not TNFR2, results in enhanced TNFR2 ectodomain shedding in neutrophils [11,115]. Therefore, sTNFR2 can reflect previous or ongoing activation of TNFR2 and/or TNFR1. sTNFR2 released from activated Tregs can scavenge TNF, and thereby prevents the inflammatory effects of TNF and can maintain the immunosuppressive function of Tregs against Teffs [116]. Additionally, high serum-levels of sTNFR2 serve as prognostic marker with poor clinical outcome in various cancer types [117,118,119,120].

Similar to Tregs, MDSCs can promote tumor immune escape. It is therefore not surprising that many tumors also show an increased number of MDSCs [78]. MDSCs suppress anti-tumor immune responses through several mechanisms; for example, by producing NO and ROS, depletion of cysteine or release of IL10 and TGFβ [121]. Tumor cells themselves release many of the stimulating factors of myelopoiesis, such as granulocyte-macrophage colony-stimulating factor (GM-CSF), stem-cell factor and vascular endothelial growth factor (VEGF), all of which promote the development of MDSCs [122,123,124]. In 2012, Hu et al. demonstrated that memTNF promotes tumor growth, tumor progression and angiogenesis [81]. This came along with increasing numbers of MDSCs and Tregs in the TME, with poor lymphocyte infiltration and higher levels of NO, IL10 and TGFβ. Vice versa, a tumor model missing memTNF expression showed higher numbers of tumor-infiltrating lymphocytes and reduced accumulation of MDSCs [81]. Studies with wildtype and TNFR2-deficient MDSCs showed furthermore that memTNF via TNFR2 upregulates CXCR4 expression, and thereby enables chemotactic migration of MDSCs to the tumor microenvironment [125].

Metastasis, and thus the successful colonization of tumor cells at distant sites and their subsequent adaption and growth, is pathophysiological, and typically the most life-threatening step in tumor development [126]. An early event when tumor cells have invaded the liver can be the expression of TNF [127]. In a study investigating the role of TNF in liver metastasis with the help of TNFR1 and TNFR2 knockout mice, it was found that TNFR2 deficiency, but not lack of TNFR1, results in a significant reduction in liver metastasis in colon and lung carcinoma tumor model [128]. Furthermore, TNFR2 deficiency and reduced metastasis were correlated with decreased amounts of CD11b^+^ and Gr1^+^ MDSCs and less Tregs in metastases site [128]. Remarkably, the pro-metastatic activity of TNFR2, which is evident from this study, turned out to be female specific, as in male mice, loss of TNFR2 did not result in a reduced amount of liver metastasis [129]. Furthermore, these findings correlated with the estrogen level and the finding that estrogen induces TNFR2 expression in isolated splenocytes [129]. Thus, reduced numbers of liver metastases were detected in ovariectomized female C57BL/6 mice, and this could be reverted by estradiol reconstitution. Moreover, Tregs and MDSCs of ovariectomized mice show reduced TNFR2 expression and reduced T-cell suppressive activity [129]. It is worth mentioning that TNFR2 might not only promote metastasis through the stimulating effect of TNFR2 on immune suppressive cells, but possibly also by affecting the activity of NK cells. It has been found that the expression of the immunosuppressive NKp30C isoform in gastrointestinal stromal tumors is tightly associated with autocrine TNF/TNFR2-induced upregulation of TRAF1 and cIAP2, along with downregulation of the activating NK cell-receptor NKp46 [130]. However, the possible causal relationship of this correlation remains to be clarified.

### 4.2. TNFR2 as An Oncogene

Aberrant expression of TNFR2 in certain cancers is accompanied by poor survival prognosis. As already discussed above, sTNFR2 levels are associated with poor survival prognosis in multiple myeloma, Hodgkin’s lymphoma, colorectal cancer, cutaneous non-Hodgkin’s disease and ovarian cancer [117,118,119,120,131]. Similarly, increased TNFR2 expression correlating with tumor size and clinical stage was reported for breast cancer [132]. High TNFR2 expression has also been found to be associated with shorter survival of patients suffering from non-small cell lung cancer (NSCLC) [133] and recently a systemic screening of almost 800 tumor cell lines derived of different cancer types revealed widespread TNFR2 expression, not unexpected especially in hematopoietic and lymphoid cancer cell lines [89]. These clinical data suggest that TNFR2 might also show pro-tumoral behavior within the tumor cell itself.

The pro-tumoral activities of TNFR2 must not necessarily originate from its role in specifying the properties of the TME but can also be based directly on oncogenic TNFR2 activities. Early on, it was found in a mouse model of DMBA/TPA-induced skin cancerogenesis that, particularly at the beginning of skin-tumor development, TNF is a major factor driving dermal inflammation and keratinocyte hyperproliferation [134,135]. Follow-up studies with TNFR1 and TNFR2 knockout mice revealed a crucial role of TNFR1 in skin cancerogenesis in this model, but also showed a significant contribution of TNFR2. These in vivo data correlated with in vitro studies showing a dominant role of TNFR1 in TNF-induced expression of GM-CSF and MMP9 in primary keratinocytes, but also a clear reduction in the inducibility of these factors in the absence of TNFR2 [136].

In a rare type of cutaneous T-cell lymphoma with poor clinical outcome, the Sezary Syndrome (SS), TNFR2 point mutations and aberrant gene duplications covering the TNFR2 gene locus lead to increased proliferation and cell growth due to enhanced NFκB signaling [137]. Likewise, TNFR2 promotes cancer-cell proliferation in colorectal cancer (CRC). Analysis of CRC tissue through immunohistochemistry revealed that high TNFR2 expression in the cancer cells is associated with higher expression of the proliferation marker Ki67 [138]. Furthermore, stable expression of TNFR2 in the SW1116 cell line enhanced proliferation in this study, while silencing of TNFR2 in the HT29 cell line reduced proliferation [138]. TNFR2 upregulation lead to enhanced protein kinase B (AKT) activity, suggesting that TNFR2 drives CRC progression via the phosphoinositide 3-kinase/AKT signaling pathway [138]. TNFR2 upregulation in colonic epithelial cells has also been reported in TNF-dependent cancer development associated with AOM/DSS-induced colitis [139]. Furthermore, follow-up studies gave evidence that TNFR2-induced MLCK expression disrupted tight junctions and thereby promoted cancerogenesis by triggering epithelial cell proliferation through luminal bacteria-induced inflammation [140].

It is well known that hepatic progenitor cells (HPCs) show abnormal compensatory proliferation in conditions of inflammation and tissue damage [141]. Indeed, TNF, presumably acting via TNFR2 and STAT3, has been identified as a crucial factor of HPC activation in a DEN-induced model of hepatocellular carcinoma [142]. Moreover, TNFR2 was recently identified as a driver of primary liver cancer (PLC) under involvement of YAP (Yes-associated protein) [143]. TNF can activate HPCs, but only TNF-TNFR2 interaction on HPCs leads to malignant transformation driving liver tumorigeneses. In this process, TNFR2 stimulation activates YAP signaling [143], an important component of the Hippo pathway, which regulates organ size and tumorigeneses [144]. Phosphorylated YAP is restrained in the cytoplasm and subsequently degraded. However, dephosphorylated YAP can enter the nucleus and, together with transcription factors of the TEAD (transcriptional-enhanced associate domain) transcription factor family, regulates the expression of several target genes [144]. The TNFR2 stimulation on HPCs promotes YAP signaling through the direct binding of heterogeneous nuclear ribonuclear protein K (hnRNPK), thereby stabilizing YAP on target-gene promoters and promoting malignant transformation of HPCs. Furthermore, single-cell RNA sequencing showed that the expression levels of TNFR2, YAP and hnRNPK in PLCs are enhanced and associated with a poor survival prognosis. All these findings point to the TNFR2-YAP axis as an important driver of HPC progression to PLC [143].

TNF has also been identified as a promoter of tumorigenesis in mice expressing the neu/erbB2 oncogene in the mammary epithelium or under control of the murine mammary tumor virus long terminal repeat [145,146,147]. Moreover, as mentioned above, TNFR2 expression has been positively associated with reduced overall survival time and disease-free survival in breast cancer patients [132]. Again, one pro-tumoral mechanism of TNF/TNFR2 signaling in breast cancer cells seems to be activation of the Akt pathway, this time resulting in upregulation of the DNA damage-repair protein poly(ADP-ribose) polymerase [148]. An association of TNF and the Hippo signaling pathway, as discussed above for liver cancer, has also been noted in breast-cancer-cell migration [149]. However, the role of TNFR2 has not been investigated in this study. Notably, there is evidence that TNFR2 can also act as a tumor suppressor in breast cancer. It has been observed that the loss of one TNFR2 allele in breast-cancer-prone MMTV-Wnt1 mice results in ductal hyperplasia in the mammary gland, higher numbers of mammary epithelial stem cells and last but not least, in an increased incidence of tumors with an aggressive metastatic phenotype [150]. The underlying mechanisms are poorly investigated but may involve autocrine TNF production [150].

In summation, TNFR2 can promote tumor development and metastasis but is also able to elicit anti-tumoral activities (Table 1). This makes TNFR2 a promising tumor therapeutic target which, dependent on the concrete circumstances, might be addressed by agonists or antagonists.

## 5. TNFR2 as a Therapeutic Target in Cancer

The different functions of TNFR2 as an oncogene and as an immune regulator of the TME open up to opposing strategies to target TNFR2 for cancer therapy, such as, on the one hand, inhibition of TNFR2 activities in immunosuppressive cells or TNFR2^+^ cancer cells or the TNFR2-dependent killing of these cells, and on the other hand, the stimulation of TNFR2 to exploit its proinflammatory activities, especially its co-stimulatory effect on CD8^+^ T-cells.

### 5.1. Antagonistic Anti-TNFR2 Antibodies in Preclinical Tumor Models

Antagonistic anti-TNFR2 antibodies have been frequently used in vitro in the TNF field to differentiate between TNFR1- and TNFR2-mediated effects but there is also some in vivo experience with antagonistic TNFR2 antibodies. Antagonistic anti-TNFR2 antibodies are characterized by binding to TNFR2 in a way that prevents ligand binding. Importantly, anti-TNFR2 antibodies typically elicit strong agonism when presented in FcγR-bound form, irrespective of the epitope recognized [151]. It is therefore crucial for potential in vivo applications of ligand blocking anti-TNFR2 antibodies as TNFR2 antagonists to use them in an IgG isotype lacking/minimizing FcγR binding.

In 2017, Torrey et al. described two dominant antagonistic TNFR2 antibodies, whereby dominant describes the ability of these antibodies to inhibit Treg proliferation and TNFR2 shedding in an FcγR-independent manner and to kill OVCAR3 cells after prolonged incubation [101]. Both of these antagonistic antibodies recognize a similar region within TNFR2, and Torrey et al. speculated that these antibodies “fix” ligand-free TNFR2 dimers in an inactive state with inaccessible TNF-binding sites. The strongest effects with these antibodies were found when Tregs were isolated directly from ovarian cancers in comparison to Tregs from peripheral blood of ovarian cancer patients or healthy donors, indicating some kind of tumor Treg preference. This tumor preference might be explained by the fact that in tumors, the proportion of TNFR2^+^ Tregs is unusually high compared with healthy tissue, indicating more available targets for TNFR2-specific antibodies or by “priming” events occurring in the tumor microenvironment, making Tregs more dependent on TNFR2 signaling. One of these TNFR2-specific antibodies was also tested for its ability to inhibit CD4 T-cells of Sézary syndrome (SS) patients and various tumor cell lines. TNFR2^+^ CD4 cancer T-cell frequencies were lowered in response to the treatment and the ratio of Tregs/Teffs was restored compared to healthy controls, confirming the potential of antagonistic TNFR2 antibodies in cancer therapy [152]. An IgG2 mutant (C232S and S233S) of this antibody with a stabilized hinge region and broad separation of antibody arms was found to be particularly active in inhibition of cell growth of cancer cell lines with high TNFR2 expression [89]. Unfortunately, the mechanisms and mode of growth inhibition/cell death of the anti-TNFR2 antibody treated cells were not further investigated in the above-mentioned studies. It is thus unclear how inhibition of TNFR2 signaling translates at the molecular level in these studies in growth inhibition or cell death.

The activation of dendritic cells (DCs) is a goal of many immunotherapies. The stimulation of Toll-like receptor 9 (TLR9) on plasmacytoid DCs (pDCs) with CpG oligodeoxynucleotides (ODNs) can induce anti-tumor response in mouse models [153]. However, there is also evidence from ex vivo experiments with human pDCs that this kind of treatment can also trigger the induction of immunosuppressive Tregs [154,155]. The latter finding resulted in initial studies with the aim to boost the immunostimulatory effect of CpG treatment with the help of antagonistic TNFR2 antibodies (anti-mouse TNFR2 murine IgG1 M831 from Amgen; anti-mouse TNFR2 hamster IgG TR75-54.7, commercially available). In the CT26 mouse model of colon cancer, the administration of the anti-TNFR2 antibody M861 resulted in CpG-treated mice in a decreased number of TNFR2^+^ Tregs, increasing amounts of tumor-infiltrating CD8^+^ Teffs and higher survival rates [155]. In addition, after successful therapy, the mice were resistant to repopulation with the same tumor, but not to 4T1 breast cancer cells [155]. One has, however, to take care in the interpretation of these results with respect to the mode of action of the putative antagonistic anti-TNFR2 antibodies. Since these antibodies carries no mutations silencing FcγR interaction, it cannot be ruled out that FcγR-mediated activities or agonism of FcγR-bound antibody molecules contributed to the observed treatment effects. Indeed, it has been demonstrated elsewhere that the antagonistic hamster anti-mouse TNFR2 antibody TR75-54.7 acts as a TNFR2 agonist after cross-linking in vitro [156] and binds to murine FcγRs II and III [157]. These considerations are also applicable for BI-1808, a fully human antagonistic ligand blocking anti-TNFR2 IgG1 antibody from BioInvent International AB [158], which is currently under investigation in a first clinical trial in patients with advanced solid tumors and cutaneous T-cell lymphoma (clinicaltrials.gov identifier NCT04752826).

### 5.2. Anti-TNFR2 Antibody Agonism in Preclinical Tumor Models

In 2019, the mouse TNFR2 specific Y9 antibody has been described [159]. The antibody was generated as mouse IgG2a isotype, has TNF-blocking capabilities and binds to CRD1 of TNFR2. Y9 showed anti-tumor responses in syngeneic mouse models with several mouse cancer cell-lines indicated by CD4^+^ and CD8^+^ T-cell expansion, with increased functionality of CD8^+^ Teffs, downregulation of memTNF and reduced tumor size. These effects were not accompanied by Treg depletion, but FcγR interaction was necessary, suggesting that the agonism of FcγR-bound antibody molecules is important [159]. Similar results were obtained with agonistic TNFR2 antibodies in a CT26 syngeneic tumor model [157]. TNFR2 can also be efficiently and selectively engaged using recombinant oligomeric TNFR2-specific TNF mutants [3], but these reagents have not been tested yet in cancer models.

In addition to the anti-TNFR2 antibodies discussed above, several more are currently in preclinical/commercial development (Table 2). Unfortunately, only a few peer-reviewed publications are available for most of these antibodies. Most often, these antibodies were only briefly presented at conferences, so a generalized consideration which strategy for TNFR2 targeting is best suited for which disease type is currently not possible.

### 5.3. TNFR2 Targeting and Immune Checkpoint Blockade

To further improve clinical outcome, combination therapies of agonistic TNFR2-targeting reagents with other anti-cancer drugs or biologicals are considered. An obvious possible approach is the combination of TNFR2 agonists with immune checkpoint blockade (ICB). Over the last few decades, immunotherapies targeting immune checkpoint molecules such as CTLA-4 (e.g., Ipilimumab) or PDL1 (e.g., Avelumab) have shown great efficacy. Nevertheless, these therapies can promote undesirable autoimmune effects and only 30 % of treated patients show a good anti-tumor response. In the above-cited study by Tam et al. [159], the combination of the anti-TNFR2 antibody Y9 with its assumed FcγR-dependent agonism and a PDL1 blocking antibody resulted in superior anti-tumor activity. Furthermore, case and colleagues combined PD1 blockade with one of the antagonistic TNFR2 antibodies described by [101] in mouse colon cancer models (CT26 and MC38). The combination therapy exceeded the efficacy of the two corresponding monotherapies. The combination therapy led to a reduction in immunosuppressive Tregs and a normalization of the Treg/Teff ratio. Moreover, the best results were obtained when both blocking antibodies were administered simultaneously and not administered at different time-points [168]. Since the anti-TNFR2 antibody used was not silent for FcγR binding, the mode of action of the anti-TNFR2 antibody (ADCC of Tregs, TNFR2 blockade, TNFR2 engagement by FcγR-bound antibody molecules) is not fully clear in this study. Indeed, ADCC-mediated depletion of Tregs, along with enhanced activity of CD8+ T-cells, has been identified as the mode of action of the anti-TNFR2 antibody TY101 in a syngeneic murine tumor model [169].

In a genome-wide crispr-Cas9 screen, Vredevoogd et al. [170] identified TRAF2, cIAP1 and cIAP2, but also several other components of the TNF-stimulated TNFR1/TNFR2 signaling network as major factors regulating the T-cell sensitivity of tumors. Follow-up analysis revealed (i) that TNF expression in untreated cancers does not correlate with survival, but also revealed a positive correlation with responders of anti-PD1 therapy; (ii) that TRAF2 deficiency sensitizes tumor cells for the cytotoxic action of CD8^+^ T cell-derived TNF and (iii) that TWEAK-induced Fn14 activation sensitizes cancer cells for killing by CD8^+^ T cell-derived TNF-induced cell death by virtue of depletion of the available pool of TRAF2-cIAP1/2 complexes, thus by similar mechanisms as already described for TNFR2 (see also 2.) [170]. Thus, in the case of TNFR2-expressing tumor cells, TNFR2 agonists may not only improve TNF release by CD8^+^ T-cells, but may also sensitize the tumor cells for TNF-induced cell death. Indeed, TNFR2 has been identified as a molecular marker for ALL patients who have a good chance to respond to cIAP1/2 antagonists, such as birinapant, and this is associated with TNFR2-dependent recruitment of RIPK1 to TNFR1 and RIPK1-mediated cell death [171]. In summation, these recent findings prompt testing therapeutic strategies combining anti-PD1 checkpoint blockade with Fn14 activation to sensitize cancer cells for TNF killing and/or TNFR2 activation to enhance CD8^+^ T-cell activation and TNF release.

It is worth mentioning that there is evidence that some immunomodulatory drugs, such as lenalidomide, but also chemotherapeutic drugs, such as cyclophosphamide, azacitidine, elicit, a least in part, their therapeutic effects through the suppression of Tregs or modulation of TNFR2 expression on Tregs [172,173,174].

## 6. Conclusions

The numerous preclinical studies with TNFR2-deficient mice and TNFR2-targeting biologicals argue for an exceptionally good tolerability of both TNFR2 antagonists and TNFR2 agonists, with high antitumoral effectiveness at the same time. The high hopes set on TNFR2 as a therapeutic target are also reflected by the fact that several companies have started clinical/commercial development of anti-TNFR2 antibodies in recent years (Table 2). Clinical studies must now show whether the good tolerability profile observed in animal models is also evident in clinical practice, particularly in patients who have comorbidities or are co-administered with other anticancer drugs. In view of the strongly diverging possibilities of targeting TNFR2 (activation or inhibition of TNFR2 signaling, destruction of TNFR2^+^ cells), which all showed promising results in preclinical models, a major challenge is certainly the question of on which basis or with which rationale cancer patients can be selected for a defined TNFR2 targeting strategy (TNFR2 blockade, TNFR2 engagement, ADCC) in clinical trials.

## Figures and Tables

**Figure 1 cancers-14-02603-f001:**
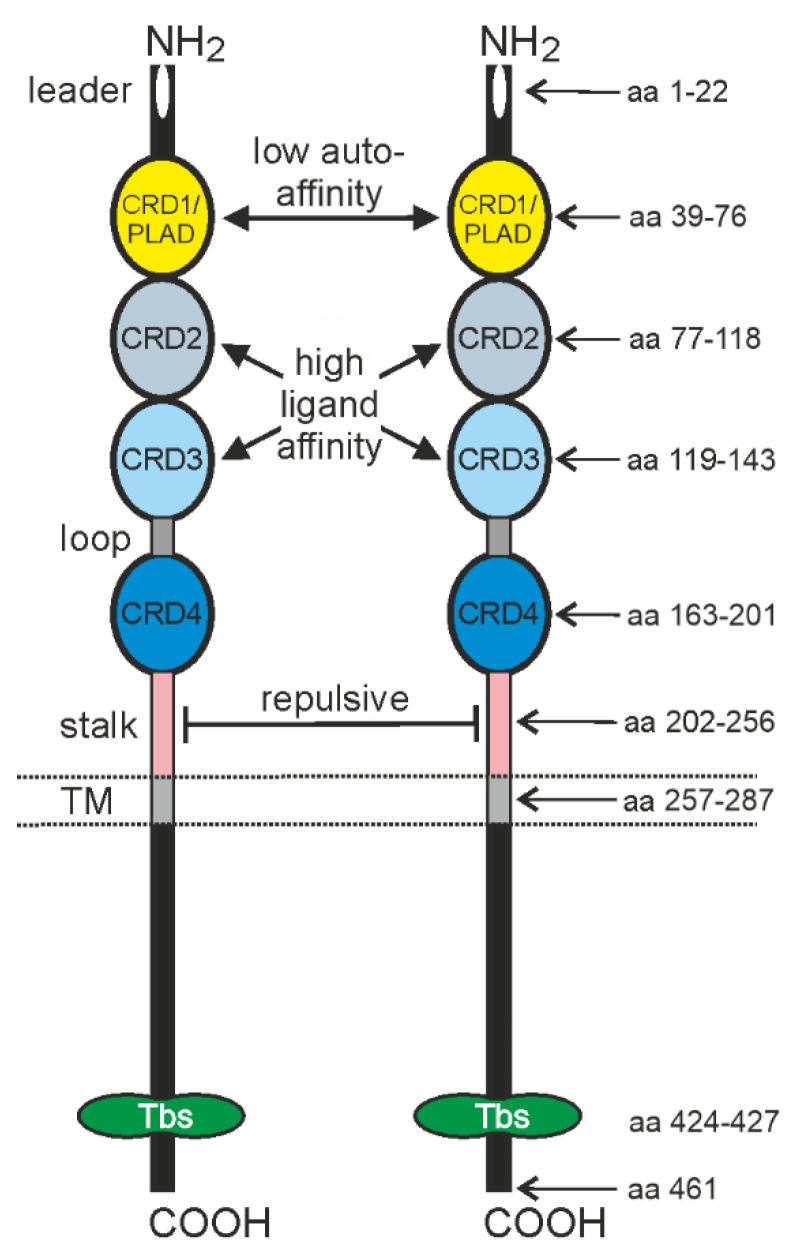
Domain architecture of TNFR2. CRD1–CRD4 of TNFR2 define the affiliation to the TNFRSF. Structurally poorly defined parts of CRD1, designated as pre-ligand binding assembly domain (PLAD), mediate low-affinity TNFR2 self-assembly in the absence of TNF. CRD2 and CRD3 mediate high-affinity binding of TNF while the function of CRD4 is largely unclear. The stalk region has repulsive features and antagonize PLAD action. TRAF2 binding site (Tbs) indicates a short amino acid motif interacting with the C-TRAF domain of a TRAF2 protomer.

**Figure 2 cancers-14-02603-f002:**
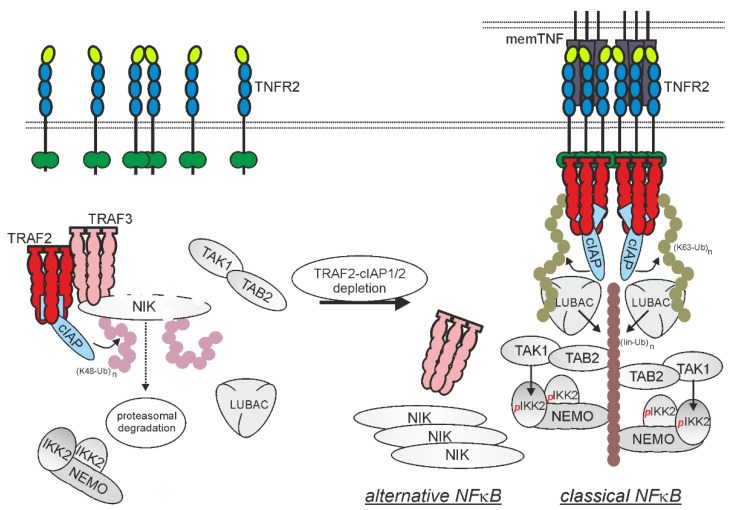
Membrane TNF-induced NFκB signaling. In the absence of memTNF, TRAF3-cIAP1/2-NIK complexes interact with TRAF2-cIAP1/2 complexes in the cytoplasm resulting in cIAP1/2-mediated K48 ubiquitination of NIK and proteasomal degradation of NIK (left panel). In memTNF-stimulated cells trimeric memTNF-TNFR2 complexes recruit TRAF2-cIAP1/2 complex and thus reduces their availability for triggering NIK degradation. As a consequence, NIK accumulates and drive alternative NFκB signaling. The trimeric memTNF-TNFR2 complexes aggregates secondarily to clusters enabling proximity of two or more TRAF2-cIAP1/2 complexes. cIAP transactivation then leads to K63-ubiquitination of TRAF2 generating docking sites for classical NFκB signaling-stimulating kinases.

**Figure 3 cancers-14-02603-f003:**
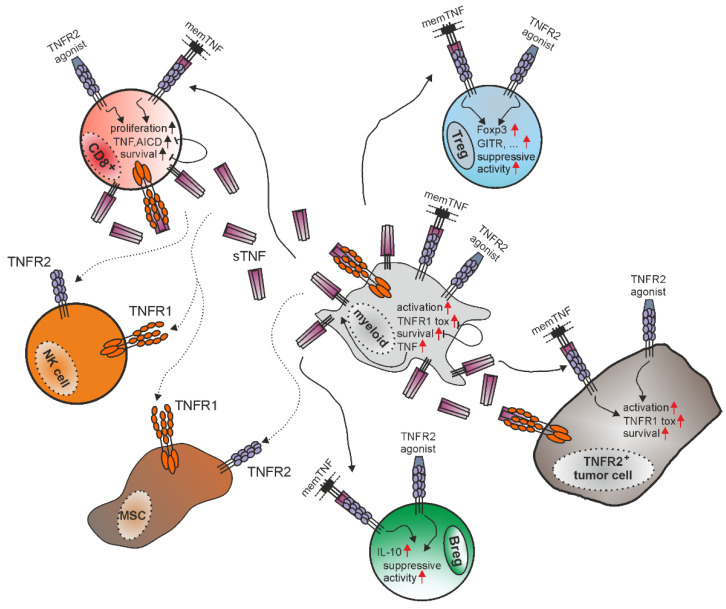
The biology of TNFR2, e.g., in the tumor microenvironment, is determined by the mutual influence of various TNF and TNF receptor expressing cell-types and a complex intracellular signaling network of the two TNF receptors. TNFR2 is expressed by various immune cells with partly opposing functions, e.g., immune suppressive Tregs but also CD8^+^ T-cells and various types of myeloid cells. Due to its ability to modulate the quality of TNFR1 signaling (inflammation versus cell death), TNFR2 furthermore elicits, in a context-dependent manner, quite contrasting effects on the same type of immune cell (survival versus death, e.g., in CD8^+^ T-cells and macrophages). In fact, for Tregs, macrophages and CD8^+^ T-cells, comprehensive literature is available describing partly counteracting activities of TNFR2 at the cellular level. This also applies for other cell types in which TNFR2 activities are less well investigated, such as, for example, NK cells and mesenchymal stem cells (MSCs) or even TNFR2-expressing tumor cells. For further details, see main text.

**Table 1 cancers-14-02603-t001:** Tumor-related functions/activities of TNFR2.

TNFR2 Expressing Cell Type	Effect on Cell	Effect on Tumor	Ref.
MDSC	CXCR4 expression and tumor recruitment Increase in suppressive activity	Immune escape Metastasis	[81,89,125,128]
Treg	Expansion Phenotypic stability Increase in suppressive activity	Immune escape Metastasis	[59,98]
NK cell	Inhibition of NKp46 expression	Metastasis	[130]
Tumor cell	Survival signaling Proliferation	Tumor progression Malignant transformation	[105,106,107,137,143,145,146,147]
Tumor cell	Cell-death sensitization	Anti-tumoral	[110,111]

**Table 2 cancers-14-02603-t002:** Translational TNFR2-targeting antibodies.

Antibody	Company	Mode of Action	Status	Relevant Patent	Ref.
AN3025	Adlai Norty USA Inc.	Antagonist ADCC	Preclinical	-	[160]
APX601	Apexigen Inc.	Antagonist ADCC	Preclinical	WO2021/055253A2	[161,162]
Bi-1808	BioInvent International AB	Antagonist	NCT04752826	WO2020/089474A1	[158]
Bi-1910	BioInvent International AB	Agonist	Preclinical	WO2020/089473A2	[163]
BITR2101	Boston Immune Technologies & Therapeutic Inc.	-	Preclinical	-	-
HFB200301	HiFiBiO Therapeutics	Agonist	NCT05238883	WO2021/141907A1	[164]
LBL-019	Nanijng Leads Biolabs Co Ltd.	-	NCT05223231	WO2021/249542A1	-
NBL-020	NovaRock Biotherapeutics	FcγR dependent	Preclinical	-	[165]
MM-401	Merrimack Pharmaceuticals	Agonist ADCC	Preclinical	WO2020/180712A1 WO2020/061210A1	[166,167]
SIM0235	Simcere	Antagonist ADCC	Preclinical	WO2021/023089	-

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
