# Peer review of "Tumor Necrosis Factor Receptor 2 (TNFR2): An Emerging Target in Cancer Therapy"

_cancers, 2022, doi:10.3390/cancers14112603_

Round 1

Reviewer 1 Report

Medler and colleagues are presenting a review article on the emerging role of Tumor Necrosis Factor Receptor 2 in cancer research as a potential target for therapeutics. The authors discuss the preclinical studies using both agonist and antagonist molecules of TNFR2, highlighting the fact that some antagonist antibody against TNFR2 may exert agonistic functions. This is a very interesting review, well written.  Please see below for some minor comments.

Figure 1. legend and Figure 1.  

Line 48. "...CRD2 and CRD..." should read: ...CRD2 and CRD3...

TRAF2 binding site (Tbs) is not readable on the dark green bubble. The authors may want to use white font.

Figure 3.

CD8+ cells: it is very hard to read what is written inside.

On the MSC cell the authors label TNFR1, they should also label TNFR2. Also, please add the acronym in the legend. Is the myeloid cell represented in the figure a MDSC? If so the authors should label the cell as such.

In the legend, the authors mention that “TNFR2 activities are less well investigated such as for NK cells and mesenchymal stem cells…”, but there are some recent studies examining the role of TNFR2 in these cell types that would be worth mentioning in the main text (Ivagnes et al. Oncoimmunology 2018: 7(12): e1386826; Beldi et al. Front. Cell Dev. Biol. 2020: 8: 596831).

Line 101. “The latter K63 ubiquitinate components of…” Please clarify. Is this cIAP2 only that ubiquitinates the components of the TNFR2 signaling complex? And which components are ubiquitinated?

Line 208. The statement: "...increase in freely available cytoplasmic TRAF2-cIAP1/2 complexes in TNFR2 stimulated cells as discussed above." seems opposite to the one line 136: "If there is reduced availability of cytoplasmic TRAF2-cIAP1/2 complexes, e.g. due to stimulation of TNFR2..." Please clarify.

Line 286. "...leukomogenisis." should read: ...leukemogenesis.

Line 320. remove 2012

Line 454-464. This paragraph needs to be reworded for clarity. It contains run on sentences.

Line 495."In the above study by [153],..." Please indicate the name of the author.

Line 480. "...with serval..." should read: ...with several...

Line 527. "...an exceptional good..." should read: ...an exceptionally good...

Reviewer 2 Report

The review article by Medler et al. focused on the TNFR2 signaling pathway and its immunoregulatory functions and discussed the clinical role of TNFR2 in cancer therapy. The article is well organized and based on up-to-date literatures. The authors provide a comprehensive review of the role of TNFR2 in regulating cancer development. The discussion of TNFR2 as a therapeutic target would help gain understanding of the mechanism through which the agonists and antagonists of TNFR2 elicit antitumoral activity. However, there are some issues need to be addressed to improve the manuscript:

  1. lines 14, 19 “CD+8, ofTNFR2” need to be corrected.
  2. line 30 “but elicits also strong” line 38 “CRD1 is located” confusing.
  3. line 54 “LTα2ß” Greek letters should be used properly.
  4. line 88-93 should add references.
  5. line 177-121 should add reference.
  6. line 320 “2012” should be deleted.

Reviewer 3 Report

Reviewer comments:

This review covers the areas regarding TNFR2 signaling and the immunoregulatory functions related to preclinical findings and insights gained with TNFR2 agonists and antagonists. The center of this article is to address which TNFR2-associated molecular and cellular mechanisms underlie the observed antitumoral activities of TNFR2 agonists and antagonists.

  • This manuscript is for the most part well written with substantial relevant references and presented clearly. Discussion was well discussed according to the evidence provided.

Minor comments:

  • Please include the tables for the sections “4.TNFR2 in Cancer” and “5. TNFR2 as a Therapeutic Target in Cancer”.

Reviewer 4 Report

I have few comments to make:

  1. The authors are suggested to include some important anti TNFR2 antagonist antibodies used in cancer treatment such as APX601, TY101, M861, etc with types of conditions they were used for and what was the outcome of that study. 
  2. The authors are suggested to include more important TNFR2 blocking agents other than potential antagonist and agonists along with their characteristics and application such as Azacitidine and lenalidomide, Cyclophosphamide, Thalidomide and fludarabine etc. 
  3. For better understanding and relevance of context, authors are advised to elaborate the role of TNFR2 in immune regulation for example, in regulatory T-cells, TNFR-2 enhances cell proliferation and stability through signaling pathways such as IKK/NF-kB, mTOR, and MAPK; in NK cells, TNFR2 activates the BIRC3/TRAF1 signaling pathway etc. 
  4. The authors are suggested to include The clinical progress of TNFR2-Targeting Treatment antibody research and development along with the company name, their country, clinical phase, functions etc such as BITR2101, APX601, MM-401 
  5. What are the hurdles/obstacles in targeting TNFR2 in cancer therapy and how can they be overcome?
